# Retrospective Analysis of Subsolid Nodules' Frequency Using Chest Computed Tomography Detection in an Outpatient Population

Ana Paula Zanardo [1,2,*], Vicente Bohrer Brentano [1], Rafael Domingos Grando [1,2], Rafael Ramos Rambo [1,2], Felipe Teixeira Hertz [1], Luis Carlos Anflor Junior [1], Jonatas Favero Prietto Dos Santos [1,2], Gabriela Schneider Galvao [1,2] and Cristiano Feijo Andrade [2,3]

[1]  Hospital Moinhos de Vento, Porto Alegre 90560-030, Brazil; vbobrentano@hcpa.edu.br (V.B.B.); rdgrando@gmail.com (R.D.G.); rafael.rambo.1989@gmail.com (R.R.R.); fthertz@gmail.com (F.T.H.); anflorjunior0807@gmail.com (L.C.A.J.); jonatas.faverop@gmail.com (J.F.P.D.S.); gabisg@gmail.com (G.S.G.)
[2]  Postgraduate Course in Pulmonology Universidade Federal do Rio Grande do Sul, Porto Alegre 90035-003, Brazil; cfandrade@hcpa.edu.br
[3]  Hospital de Clínicas de Porto Alegre, Porto Alegre 90035-903, Brazil
*   Correspondence: zanardoap@yahoo.com.br

**Abstract:** Introduction: The study was designed to evaluate the frequency of detection and the characteristics of subsolid nodules (SSNs) in outpatients' chest computed tomography (CT) scans from a private hospital in Southern Brazil. Methods: A retrospective analysis of all chest CT scans was performed in adult patients from ambulatory care (non-lung cancer screening population) over a thirty-day period. Inclusion criteria were age > 18 years and lung-scanning protocols, including standard-dose high-resolution chest CT (HRCT), enhanced CT, CT angiography, and low-dose chest CT (LDCT). SSNs main features collected were mean diameter, number, density (pure or heterogenous ground glass nodules and part-solid), and localization. TheLungRADS system and the updated Fleischner Society's pulmonary nodules recommendations were used for categorization only for study purposes, although not specifically fitting the population. The presence of emphysema, as well as calcified and solid nodules were also addressed. Statistical analysis was performed using R software, categorial variables are shown as absolute or relative frequencies, and continuous variables as mean and interquartile ranges. Results: Chest computed tomography were performed in 756 patients during the study period (September 2019), and 650 met the inclusion criteria. The IQR for age was 53/73 years; most participants were female (58.3%) and 10.6% had subsolid nodules detected. Conclusions: The frequency of SSNs detection in patients in daily clinical practice, not related to screening populations, is not negligible. Regardless of the final etiology, follow-up is often indicated, given the likelihood of malignancy for persistent lesions.

**Keywords:** subsolid nodules; ground glass nodules; lung; computed tomography; CT; detection

## 1. Introduction

Lung cancer is the leading cause of cancer deaths worldwide, with 127,070 estimated deaths in the United States in 2023 for both sexes [1].

Persistent subsolid nodules (SSNs) are known precursors of lung cancer [2–4], and are frequently diagnosed by computed tomography (CT) as incidental lesions. They are defined as ground glass nodules (GGNs) or part-solid density nodules (PSNs) that persist in follow-up scans of at least 3–6 months. Some authors subdivide the GGNs according to their attenuation into pure or heterogeneous types [5].

Although numerous studies address the frequency and significance of persistent SSNs in lung cancer screening patients, data are still limited or controversial on SSNs' frequency in the general population [6–8].

Also, there is evidence of increasing rates of lung cancer incidentally detected in non-smokers, especially women [9], patients who do not meet the criteria for screening.

This study was motivated to evaluate how often this subtype of lung nodule is found in the usual radiological clinical practice of a non-screening population.

## 2. Materials and Methods

This is a retrospective analysis of all chest CT scans of outpatients that underwent exams at our institution in South Brazil over a thirty-day period (September 2019). The sample was chosen by convenience, and the scans were ordered by the assistant clinicians as part of the patient's usual care. The exclusion criteria were age < 18 years and non-pulmonary protocols (e.g., cardiac CT).

The study was approved by the Institutional Review Board. Informed consent was waived.

Demographic data and medical request information were collected using a patient's filled structured form.

There were multiple different indications that led the patients to the imaging department (e.g., chronic cough, chest pain, back pain, check-up, weight loss, lymphadenopathy, chronic lung diseases, rheumatic diseases, cancer staging, and follow-up).

Chest CT scans were obtained in a 16-slice scanner (Emotion 16; Siemens, Forchheim, Germany) and a 256-slice scanner (Somatom Drive 256; Siemens, Forchheim, Germany) in supine position with or without contrast at end inspiration. Image datasets were reconstructed with 1 mm slice thickness, using 0.7 mm and 0.5 mm increments, using soft tissue and sharp kernels with standard lung window settings.

All exams were reviewed by a thoracic radiologist with ten-years of experience and compared with the first report (report of the initial evaluation).

CT images were analyzed for the presence of SSNs and their features (number, size, distribution by lobes, number of involved lobes, and solid components). SSNs were classified into pure ground glass, heterogenous ground glass, and part-solid nodules based on their attenuation on CT images and measurable soft tissue components [5] (Figures 1–3).

The LungRADS system (Lung Imaging Reporting and Data System/American College of Radiology) [10] and the updated Fleischner Society's [11] pulmonary nodules recommendations were used for categorization, although not specifically fitting the population, only for study comparison purposes.

Secondly, CT images were analyzed for the presence of calcified nodules, solid non-calcified nodules (cutoff of more or less than 10 mm), and emphysema in a visual scale [12]. We used MIP (maximum intensity projection) reconstructions, but no computer-aided detection (CAD) systems.

Nodules were measured on both axes in the axial plane, lung window, to report mean nodule diameter. Multiple nodules were included.

Current smokers were defined as active smokers or those who quit within 15 years, and former smokers were defined as those who quit more than 15 years ago.

Statistical analysis: Statistical analysis was performed using R software (R version 4.2.3; R Foundation for Statistical Computing, Vienna, Austria). Categorical variables are shown as absolute (n) or relative frequencies (%). Continuous variables such as age are shown as the mean and interquartile ranges. The Kolmogorov–Smirnov test was used to verify the normal distribution of variables. Nonparametric data were analyzed by the Mann–Whitney Wilcoxon test and Pearson chi-squared with a 0.05 significance level.

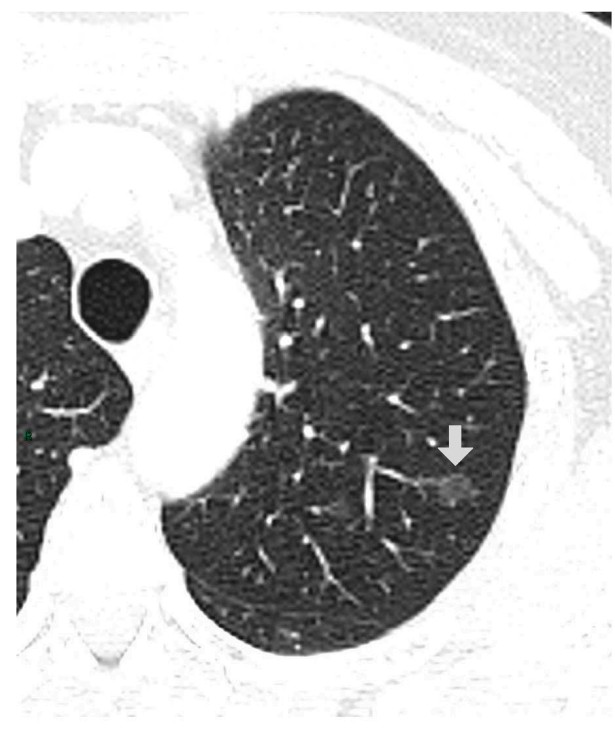

(**a**)

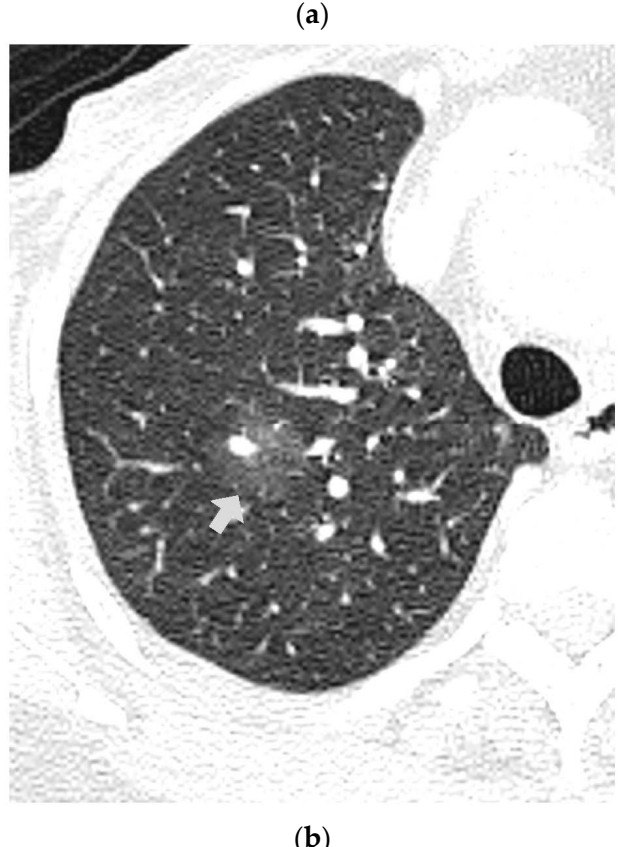

(**b**)

**Figure 1.** (**a**,**b**) Two homogenous GGNs in different patients; LUL and RUL with vascular trajectory inside.

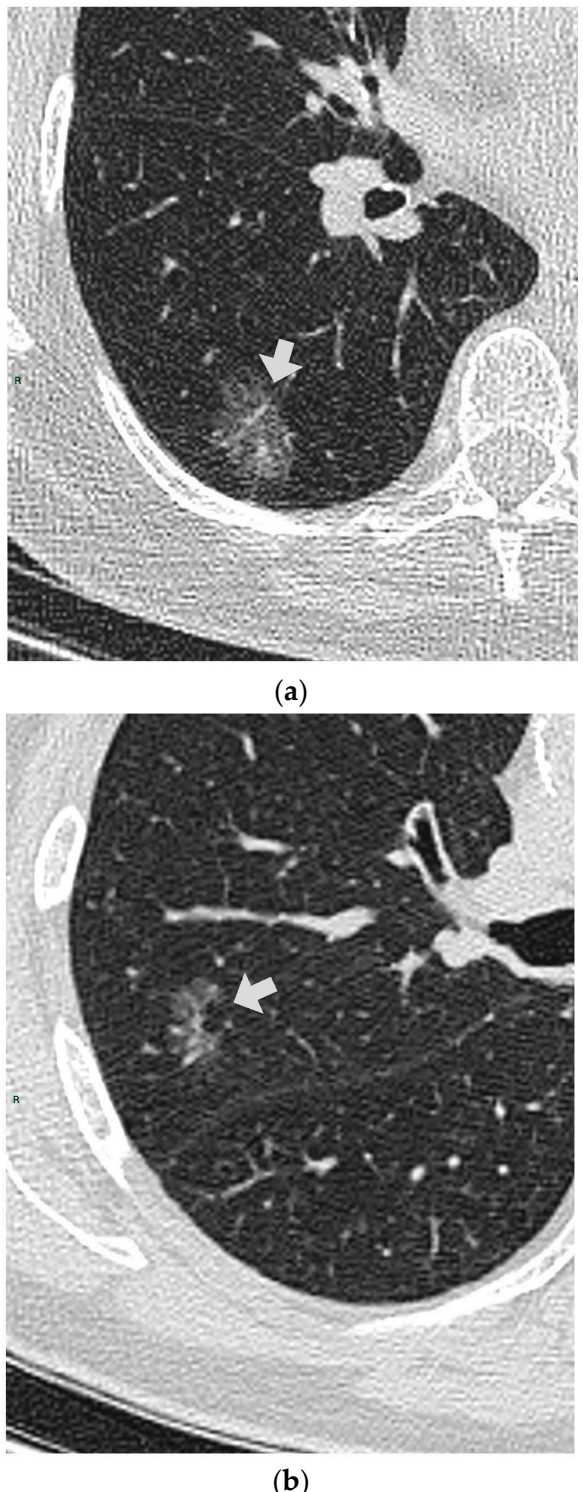

(**a**)

(**b**)

**Figure 2.** (**a**,**b**) Two heterogenous GGNs in different patients; RLL and RUL.

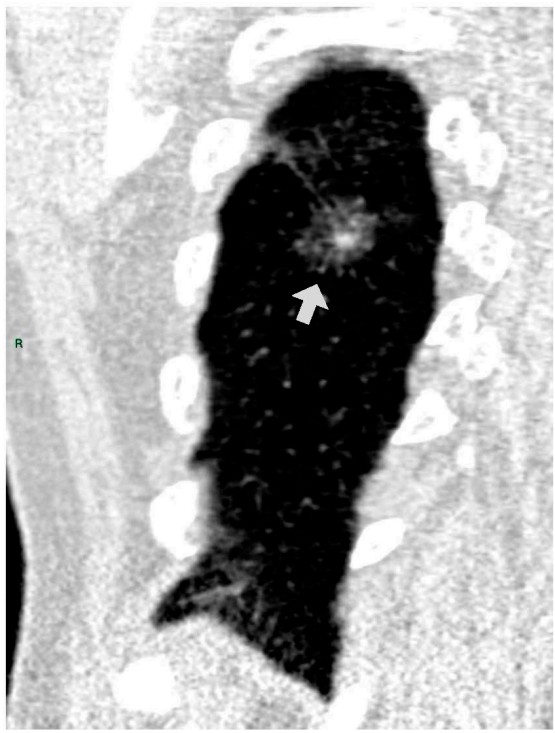

**Figure 3.** PSN in RLL (coronal reformation plane).

### 3. Results

The baseline characteristics of patients are shown in Table 1 and the study flowchart in Figure 4.

**Table 1.** Baseline characteristics of patients and in the positive SSNs group.

|  | N (%) | SSNs + (%) * | *p* |
|---|---|---|---|
| Sex |  |  | 0.008 |
| Female | 379 (58.3) | 51 (73.9) |  |
| Male | 271 (41.7) | 18 (26.1) |  |
| Total | 650 (100) | 69 (10.6) |  |
| Age |  |  |  |
| Mean (y) ± SD | 62.8 ± 14.9 | 66.8 ± 13.3 | 0.015 |
| IQR | 53/73 | 59/76 |  |
| Smoking history [a] |  |  |  |
| Former | 139 (21.4) | 15 (10.1) | 0.8 |
| Current | 149 (22.9) | 13 (9.4) |  |
| Never | 330 (50.8) | 36 (10.9) |  |
| Lacking answer | 32 (4.9) |  |  |
| Cancer history [b] |  |  |  |
| Yes | 309 (47.5) | 33 (47.8) | 1.0 |
| Lung cancer | 27 (8.7) | 5 (7.2) |  |
| Cough/dyspnea [c] |  |  |  |
| Yes | 257 (39.5) | 25 (36.2) | 0.6 |
| Emphysema [d] |  |  |  |
| Mild | 93 (14.3) | 11 (16.2) | 0.4 |
| Moderate to severe | 53 (8.2) | 8 (11.8) |  |
| No | 504 (77.5) | 50 (72) |  |

* Percentages refer to the proportion of patients with SSNs detected. [a] Current smokers refer to active smokers or those who quit within 15 years; former smokers refer to those who quit more than 15 years ago (32 patients missing data). [b] Refer to any cancer. [c] No patient referred fever in the SSNs group. [d] Emphysema graded by visual scale (reference [5]).

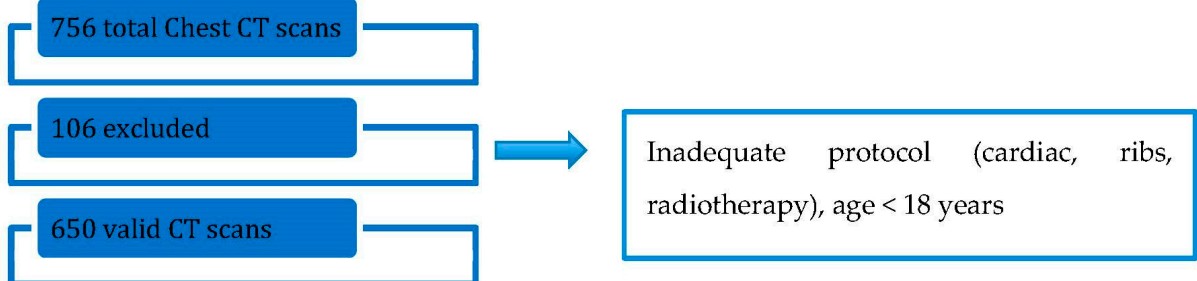

**Figure 4.** Flowchart of study inclusions and exclusions.

Total number of chest CT scans was 756, with 106 excluded (14%) and 650 that met the inclusion criteria. Excluded exams were: all cardiac protocols, radiotherapy planning, scans for ribs evaluation that miss adequate demographical data, and patients < 18 years.

Women were the majority of the population sample (58.3%).

Sex was statistically different between patients with SSNs detected (73.9% female), with a *p*-value of 0.008.

Age was higher in patients with SSNs detected (IQR 52/73 y) than without (IQR 59/76.5 y), with a *p*-value of 0.015.

The proportion of any cancer history was 47.5% (309 total, 167 women, 142 men, 27 treated lung cancer). Metastatic lung disease was present in 4.6% of the 309.

There was no statistical difference in SSNs detected among patients with and without oncologic history, respectively, 10.6% (33) and 10.5% (36), *p*-value 1.00.

Fever and/or dyspnea were referred by 2% of patients, and cough was referred by 32%. Among patients with SSNs detected, none had fever, 21 had cough, and 4 had dyspnea complaints. There was no statistical difference in SSNs detected among patients with and without referred symptoms (*p*-value 0.6).

Among the 288 patients with smoking history (active and past smokers), 10% had SSNs detected. Never smokers (330) with SSNs were 36 patients (10.9%). Smoking was not quantified.

Table 2 and Figure 5 shows the characteristics of the SSNs.

SSNs were detected in 69 patients (10.6%), in a total of 147 nodules.

Solitary nodules were found in 46 patients (67%), and multiple in 23 patients (33%), ranging from 2 to more than 10.

SSNs measuring less than 20 mm were 85% of the total. Only two nodules measured more than 20 mm (21 and 28 mm).

Pure GGNs were 87 nodules (60%) in 46 patients, heterogenous GGNs were 42 nodules (28%) in 27 patients, and part-solid nodules were 18 nodules (12%) in 15 patients.

Upper lobes were the preferred location of the most suspicious SSNs (by imaging features), accounting for 106 nodules in 44 patients, especially the right upper lobe (RUL) (62 nodules in 27 patients).

LungRADS of categories 2 and 3 nodules were classified in 61 patients, and LungRADS of category 4 were classified in 8 patients (4 multiple and 4 solitary PSNs).

Based on the Fleischner Society classification, 36 patients had solitary nodules measuring more than 6 mm (GGNs or PSNs); 23 had multiple SSNs and/or PSNs, with the majority measuring more than 6 mm; and 10 remaining patients had solitary nodules measuring less than 6 mm.

**Table 2.** Characteristics of the SSNs.

| SSNs | Nodules | Patients (%) |
|---|---|---|
| Number | 147 | 69 |
| Solitary | | 46 (67) |
| Multiple | | 23 (33) |
| | 2 | 11 |
| | 3–10 | 8 |
| | >10 | 4 |
| Size * | | |
| Up to 10 mm | | 51 |
| 11–20 mm | | 16 |
| More than 20 mm | | 2 |
| Type of SSNs [a] | | |
| Pure GGNs | 87 | 46 (60) |
| Heterogenous GGNs | 42 | 27 (28) |
| PSNs | 18 | 15 (12) |
| Location [b] | | |
| Upper lobes | 106 | 44 |
| RUL | 62 | 27 |
| LungRADS category | | |
| 2/3 | | 61 |
| 4 | | 8 |
| Fleischner Society [c] | | |
| Solitary nodules > 6 mm | | 36 |
| Multiple nodules | | 23 |
| Solitary nodules < 6 mm | | 10 |

* Considered larger size per patient. [a] Number by nodules and by patients. [b] Location of the most suspicious lesion. [c] GGNs or PSNs. Four patients in LungRADS category 4.

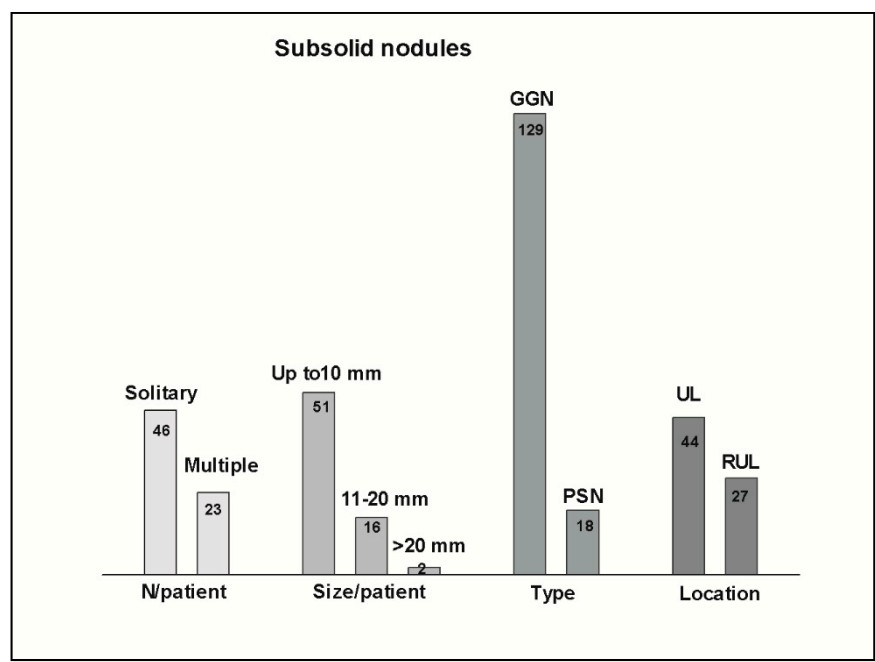

**Figure 5.** Graphic showing SSNs characteristics. N/patient refers to the number of patients with solitary or multiple lesions. Size/patient refers to the larger lesion by patient. Type refers to GGNs (pure and heterogenous) or PSNs in the total amount of nodules. Location refers to the location of the most suspicious nodule by patient in the upper lobes (ULs) and right upper lobe (RUL).

Table 3 shows the characteristics of solid and calcified nodules.

**Table 3.** Characteristics of solid and calcified nodules.

| Nodules | Patients n (%) |
|---|---|
| Solid * | 343 (52.7) |
| Size < 10 mm | (86.5) |
| Calcified | 365 (56.2) |

* Excluded patients with metastatic lung disease.

Among the 650 patients, 56.2% had calcified nodules, and 52.7% had solid nodules (metastatic lung disease was excluded). Eighty-five percent (86.5%) of solid nodules measured less than 10 mm.

Emphysema was found in 146 patients (mild 63%, moderate 26%, severe 10%). Nineteen patients with emphysema had SSNs, including the two largest nodules, and three LungRADS of category 4. Despite that, there was no statistical difference in SSNs detected among patients with and without emphysema (*p*-value 0.4).

Concerning the SSNs, the percentage of concordance between the initial report and the revision of the thoracic radiologist was 85.5% for the identification of at least one GGNs or PSNs.

Regarding the CT acquisition protocols, most exams (620) were standard-dose high-resolution chest CT (HRCT) or enhanced CT (95%), and 30 (5%) were low-dose CT (LDCT) or CT pulmonary angiography (CTPA).

Incomplete demographic data occurred in 0 to 7% of the variables collected.

## 4. Discussion

First, our study had the intent of showing the frequency of SSNs in the setting of a routine outpatient population, out of lung cancer screening programs, considering that there are controversial and limited data in this field.

Previous studies have shown that a variable amount of the SSNs is transient, ranging from 15% up to 70% [4,13–17]. Transient SSNs are more common with younger age, the presence of blood eosinophilia, and ill-defined borders, among others [16]. Persistent SSNs tends to be more common with increasing age and female sex [14,17]. In our study, the detection of SSNs was more frequent in these two subgroups, suggesting that even in patients who are not classified as "high risk" for lung cancer, the presence of these nodules must warrant extra care.

In our study, the frequency of SSNs' detection in patients with smoking history was similar to that found in never smokers, suggesting that other risk factors can contribute to its presence, especially in the particular setting of old women.

Among studies that address SSNs in routine clinical settings [6–8], there is a trend to retrospectively evaluate data related to SSNs persistence, time to progress, growth, or invasiveness. We focused on the positivity of SSNs' detection in patients out of lung cancer screening protocols and in a "single round CT", and found that SSNs' positive subgroup shared demographic characteristics with those already shown to have persistent lesions [14,17], frequently (pre)malignant.

Our study has several limitations, including a single-center outpatient sample with a high proportion of cancer history and older age. However, perspectives on population aging [18] include the growing cancer prevalence and cancer survivors.

Despite the heterogeneity of CT scan indications and acquisition protocols (both contrast and non-contrast scans were included) being considered as other methodological limitations, we can argue that all these circumstances are consistent with the routine practice of outpatient CT scan procedures. Interestingly, to the CT reading, Kyung Hee Lee and colleagues [19] found high agreement in nodule classification (subsolid and solid) between five thoracic radiologists for low-dose unenhanced CT and standard-dose enhanced CT.

Regarding the utilization of the ACR LungRADS and the Fleischner Society guidelines for pulmonary nodules, it was performed strictly for this study's comparative purposes. Although they are recommended for different clinical situations, risk stratification is some-

times unavailable for the radiologist. For nodule classification purposes, it seems that the LungRADS system has better stratified categories. The last version released in 2022 [10] removed the "risk of malignancy" column, arguing that it is highly variable and ultimately lesion-specific. The Fleischner recommendations use the 6 mm size as a cutoff to recommend follow-up for single or multiple GGNs and PSNs. Despite these differences, both systems are also based on the persistence of lesions, solid components, and growth criteria in the follow-up.

Despite the limitations inherent to the study design, our intent to estimate the frequency of SSNs in our routine outpatient population was achieved. Along with a growth in the volume of CT scans, as various studies have shown to occur both in pre- and post-COVID-19 pandemic periods [20,21], we also expect an increment in the number of patients with SSNs. As these lesions frequently have an indolent behavior [14], the duration of follow-up for these patients is becoming longer and with larger intervals.

Concerning small solid and calcified nodules, we found them to be extremely common findings in chest CT in our population, in accordance with the Brazilian lung cancer screening populations studies [22,23], which showed a high percentage of positive screenings but a much lower rate of cancer confirmation.

## 5. Conclusions

To conclude, SSNs are potentially (pre)malignant lesions, commonly found in the daily practice of thoracic radiologists. There is a trend to augment the diagnosis of these lesions along with the increasing volume of CT scans and population aging.

In our study, we found SSNs to be more frequent in older women, irrespective of smoking history. Together with other findings in the literature on SSNs and non-solid malignancies, it seems that advancing age, female sex, and upper lobe location are concerning factors for persistent (pre)malignant subsolid lesions.

Until now, the evolutive behavior of (pre)malignant SSNs is not sufficiently understood, and most patients will maintain imaging follow-up. The Fleischner Society and ACR LungRADS systems are excellent pulmonary nodules classification and management tools. Although the last was based on and is recommended for lung cancer screening populations, it has finer stratified categories for SSNs.

**Author Contributions:** Conceptualization, A.P.Z., R.D.G., J.F.P.D.S., G.S.G. and C.F.A.; methodology, A.P.Z., V.B.B., R.D.G., R.R.R., J.F.P.D.S., G.S.G. and C.F.A.; validation, A.P.Z., V.B.B., R.D.G., F.T.H., L.C.A.J., J.F.P.D.S., G.S.G. and C.F.A.; investigation, A.P.Z., V.B.B., R.D.G., R.R.R., F.T.H., L.C.A.J., J.F.P.D.S., G.S.G. and C.F.A.; data curation, A.P.Z. and C.F.A.; writing—original draft preparation, A.P.Z. and C.F.A.; writing—review and editing, A.P.Z., V.B.B., F.T.H., L.C.A.J. and C.F.A.; supervision, A.P.Z. and C.F.A. All authors have read and agreed to the published version of the manuscript.

**Funding:** This research received no external funding.

**Institutional Review Board Statement:** The study was conducted in accordance with the Declaration of Helsinki, and approved by the Institutional Review Board of Hospital Moinhos de Vento (4.260.736, 5 September 2020).

**Informed Consent Statement:** Patient consent was waived due to the retrospective observational nature of the study.

**Data Availability Statement:** Data supporting reported results can be available upon request to the authors.

**Conflicts of Interest:** The authors declare no conflict of interest.

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
