# Peer review of "Retrospective Analysis of Subsolid Nodules’ Frequency Using Chest Computed Tomography Detection in an Outpatient Population"

_tomography, doi:10.3390/tomography9040119_

Round 1

Reviewer 1 Report (Previous Reviewer 2)

Authors have addressed only some of the issues raised and the manuscript does not meet the standard criteria for publication. There are also several language issues that need to be addressed as limiting the understandability of the manuscript.

- The hospital setting, namely private, should be reported in the M&M. I do understand that “In the private health system in Brazil, the assistant clinicians order diagnostic exams as they judge necessary”, but I would think that there must be a clinical question to answer for ordering a CT (except for screening purpose). So, what kind of symptoms did the patients have? What blood tests? This aspect is relevant since GGN are commonly inflammatory in nature.

- Authors have kindly replied to my comment stating that for those patients who were administered contrast medium enhanced images were assessed. This should be reported in the MM section. Contrast medium increases the parenchymal density, leading to overestimate GGN density and extent/size. 

- With respect to cancer history, I do appreciate the reply, but this is far too generic.

- How can the number of enhanced and unenhanced CT scans not be available?? 

- Sentence “However, perspectives on population ageing (18) includes growing of the cancer prevalence and cancer survivors” is still no clear.

- Older women is too generic.

See above

Author Response

Reviewer 2 Report (Previous Reviewer 1)

Dear Authors, 

Thank you for submitting this paper, which gives the readers a clear picture, although partial, given the well exposed limitations of the studies, of the random finding of SSN in the context of routine outpatient population. This topic is particulary interesting given the current focus of the controversial issue of lung cancer screening.

Indeed, I would kindly ask you to add a brief comment on how you would place these results within the much more complex debate of lung cancer screeing.

Author Response

Thank you for reviewing our manuscript and for your comments. 

We also think that the "random SSN" findings are a particulary interesting topic, that relates with the bigger issue of lung cancer and lung cancer screening. 

Unfortunetly, we don't think that our study, given its limitations, has the power of adding real tangible news on the debate of lung cancer screening, although the results actually rise some questions about patients that seems to have augmented risk of lung cancer and are not considered for screening. Moreover, why are this patients more prone to have SSN? We don't have that answers.

Do you think that this reflections should be on the manuscript?

Thank you again. 

This manuscript is a resubmission of an earlier submission. The following is a list of the peer review reports and author responses from that submission.

Round 1

Reviewer 1 Report

Dear Authors,

Please, take these suggestions into consideration:

1)     INTRODUCTION. Lines 28-31 “Persistent...(PSN)”

Please, add more details to the description of SSN..

2)     MATERIALS AND METHODS. Lines 74-77 “CT images...components”.

Please, add images of the various sub-types of SSN.

3)     RESULTS. Lines 164-165 “Agreement...missed”

Please, add information regarding the concordance rate between the original reports of the CT scans and the revision of the thoracic radiologist as regards the detection of SSN.

Author Response

1) Persistent subsolid nodules (SSN), which are ground glass or part solid ground glass density nodules that persist in a follow up scan of at least 3-6 months, are known precursors of lung cancer, frequently diagnosed by computed tomography (CT) as incidental lesions (2,3,4). The term can be used to describe pure or heterogenous ground glass nodules (GGN) and part solid nodules (PSN) (5).

2) Images of the SSN subtypes in the attached files and in the manuscript.

3)  Regarding the SSN, concordance rate between the initial report and and the revision of the thoracic radiologist was 85,5% considering the identification of at least one GGN or PSN. This was better explained in the manuscript

The authors thank you very much for your revision and suggestions.

Reviewer 2 Report

Abstract

- Data on statistical analysis should be provided in the Methods section.

- Adequate protocol is far too generic. Details should be provided.

- Fleischner Society guidelines and Lung-RADS are mentioned in this section but not in the M&M.

- Authors should detail the characteristics of SSN they refer to.

Introduction

- Overall, paragraphs are not quite conceptually linked.

-  Forth, fifth and sixth paragraphs seem more appropriate for the discussion section. I would summarise the concept, highlighting that data on SSN frequency in the general population are still limited/controversial.  

- I would suggest moving the third paragraph to the end of the section, possibly integrated with the aim of the present study (missing).

Materials and Methods

- Chest HRCT is performed without medium contrast. Authors should specify whether the radiologist assessed unenhanced images or change the manuscript referring to chest CT instead of HRCT.

- Details on slice thickness, increment, etc should be also provided. 

- Authors should specify how many years of expertise did the radiologist have. No clear what they mean by “compared with the first report”.

- Why did these patients undergo a chest CT in the first place?

- SSN are usually distinguished into pure ground glass and part solid, as per reference 5. No clear the meaning of “heterogenous ground glass”.

- Explain how the emphysema was classified and graded.

Results

- Exclusion criteria are missing. Why were 106 CT excluded?

- As acknowledged by the Authors, Lung-RADS categories are not appropriately used here (even so, which version of Lung-RADS do Authors refer to???).

- Cancer history is far too generic. Do Authors refer to lung cancer history? Type of cancer at high risk of lung metastases?

- Which test was used to assess the agreement between the retrospective reading and the first report?

Discussion

- There is not a proper discussion of the results presented.

- Sentence “We focused on the positivity of the cross-sectional detection” is not clear.

- Sentence “Loss of data occurred from 0 to 7% of each variable” needs rewording, no clear.

- Third to last paragraph is quite inaccurate. 

- Limitations should be added.

Conclusion

- Overall, quite vague.

- Last sentence should be expanded/reworded. Lung-RADS categories can be used only in the setting of lung cancer screening and not in every day clinical practice. 

- CT images should be added.

Moderate editing needed.

Author Response

Abstract - Changes in the uploaded revised manuscript.

Introduction - Suggestions accepted and  structured in the uploaded revised manuscript.

Materials and Methods - Some more details on CT protocols were specified in the M&M section, and discussion. As we included different protocols in the intention to include all patients, we did not explicited each one. Let me know if you think that something else is recquired.

Years of expertise of the radiologist included in the manuscript. The first report refers to the initial evaluation by a radiologist at the moment of the exam. We will provide specific comparison regarding SSN description.

- Why did these patients undergo a chest CT in the first place? As a convenience sample of outpatients, the exams were ordered for multiple different reasons. In that sample, we found that around 50% of the exams were follow up of oncologic patients.

- About the reference 5. In the article, we can read as below:  "Materials and Methods: Eight facilities participated in this
study. A total of 795 patients with 1229 SSNs were assessed
for the frequency of invasive adenocarcinomas. SSNs were
classified into three categories: pure ground-glass nodules
(PGGNs), heterogeneous GGNs (HGGNs) (solid component
detected only in lung windows), and part-solid nodules."

- Explain how the emphysema was classified and graded. - Added in the revised uploaded manuscript, with reference (CT-Definable Subtypes of Chronic Obstructive Pulmonary Disease: A Statement of the Fleischner Society)

Results

- Exclusion criteria added in the paragraph.

- Lung-RADS categories: Despite Lung RADS categories derives from lung cancer screening studies, more data are showing increasing prevalences of lung adenocarcinomas in patients that do not exactly fit that righ risk population, as old women without smoking history. We decided to use the Fleischner Society and the Lung-RADS nodules categorization systems as a comparison exercise, just for study purposes, knowing that frequently radilogists doesn't have adequate clinical information of the patients in daily practice. We used the version 1.1 of the Lung Rads at the time of the study. After comparing with the late version recently released (v2022), we found no specific changes that could alter the classification of these subtype of nodules.  The question is if the Lung RADS system could not be expanded to other populations? Certainly the category changing in the classification criteria is a valid tool for a stepped management. 

- Cancer history: We refer to any cancer history, including the most common in our facility (breast, prostate, colorectal), Among 309 "oncologic" patients,  27  had a history of treated lung cancer. Lung metastases were found in a minor part of the oncologic patients (percentage included in the revised manuscript).

- Which test was used to assess the agreement between the retrospective reading and the first report? Actually, we just used a comparison of the percentage of the first reports and the retrospective reading that matched the description of at least one SSN. As the initial report was not intended to categorize the nodules, there was no posibility of a more detailed correlation.

Thank you for yor suggestions and ideas.

Reviewer 3 Report

Thank you for the opportunity to review this interesting article. This study is original and well-designed, however, some improvements should be advisable.

In line 32, when you described the reason of the study, you can stress the value of the study arguing about the necessity to evaluate the incidence of diagnosis of subsolid nodules in the standard population, out of the screening lung cancer programs, to detect the possible element to identify risk factors in this group of patients.

Please check Table 1: there are lacking numbers regarding % in the first column.

The definition of current/former smokers reported after Table 1, should be moved to Methods.

The Introduction and the discussion shoud be improved to stress the original aspects of the study and the possible pratical  value 

The English language should be revised to delete inaccuracies

Author Response

We thank you for your kind comments and excellent suggestions.

Line 32 - We did some modifications in the manuscript in the attempt to better explain the reason of the study, which is to underestand the frequency and how to adequately asses the importance of the diagnosis of subsolid nodules out of the screening lung cancer programs.

Table 1 -  Modifications already done in the manuscript.

"The Introduction and the discussion shoud be improved to stress the original aspects of the study and the possible pratical  value" - We did some improvements in the manuscript. Please let us know if it fits your expectations and ideas.

English language -  We tried to improve the hole manuscript. If you think that an editing is necessary, please let us know.

Round 2

Reviewer 2 Report

Abstract

- Low dose chest CT and high-resolution chest CT (HRCT) do not necessarily represent different protocols, since a HRCT can be obtained at a low dose radiation exposure. I would suggest referring to standard dose chest HRCT and low dose chest HRCT.

- As for my previous comment, Authors should detail the characteristics of SSN they refer to. I did not ask to provide the definition of SSN nodules, but to list/explain which characteristics of them were considered (volume? density? larger diameter?).

Introduction

SSN are usually distinguished into pure ground glass and part solid. No clear the meaning of “heterogenous/homogeneous ground glass”.

- Sentence “The term SSN can be used to describe pure or heterogenous ground glass nodules (GGN) and part solid nodules (PSN) is redundant, it should be removed.

- As for previous comment, some sentences/paragraphs seem more appropriate for the discussion section. I would summarise the concept, highlighting that data on SSN frequency in the general population are still limited/controversial rather than detailing results of previous studies.

- The aim of the present study is still not clearly stated.

Materials and Methods

- Firs sentence needs rewording. No clear the meaning of “convenience sample”.

- Sentence “The scans were ordered by assistant clinicians in their usual practice” is not clear.

- Authors should specify whether the radiologist assessed unenhanced images for those patients who were administered contrast medium.

- Why did these patients undergo a chest CT in the first place?

Results

- Cancer history is far too generic. Do Authors refer to lung cancer history? Type of cancer at high risk of lung metastases?

- Which test was used to assess the agreement between the retrospective reading and the first report? This should be reported in the statistical analysis paragraph.

- The number of enhanced and unenhanced CT scans should be detailed.

Discussion

- As far as I understand, a single time-point was assessed. As such, not clear why Authors comment on the significance of SSN persistence. 

- Sentence “However, it must be remarked that population ageing is a fact (18), as well as the tendency to increasing the incidence and diagnosis of neoplastic diseases.” needs rewording, no clear.

- Third to last paragraph still quite inaccurate.

Conclusion

- Overall, quite vague.

- No clear whether sentence ”especially in older women without specific risk factors for lung cancer and out of screening protocols” refers to this study (if so, how can this be stated given that all patients underwent their CT outside lung cancer screening programs??) or to previous literature.

- The Fleischner Society guidelines and Lung-RADS were designed for different purposed. Why do Authors compare them??

- CT findings showed on images should be indicated with arrows (or similar).

See above
